# Improved Dipole Source Localization from Simultaneous MEG-EEG Data by Combining a Global Optimization Algorithm with a Local Parameter Search: A Brain Phantom Study

**DOI:** 10.3390/bioengineering11090897

**Published:** 2024-09-06

**Authors:** Subrat Bastola, Saeed Jahromi, Rupesh Chikara, Steven M. Stufflebeam, Mark P. Ottensmeyer, Gianluca De Novi, Christos Papadelis, George Alexandrakis

**Affiliations:** 1Bioengineering Department, The University of Texas at Arlington, Arlington, TX 76019, USA; saeed.jahromi@uta.edu (S.J.); rupesh.chikara@cookchildrens.org (R.C.); christos.papadelis@cookchildrens.org (C.P.); galex@uta.edu (G.A.); 2Neuroscience Research Center, Jane and John Justin Institute for Mind Health, Cook Children’s Health Care System, Fort Worth, TX 76104, USA; 3Athinoula A. Martinos Center for Biomedical Imaging, Massachusetts General Hospital, Harvard Medical School, Boston, MA 02129, USA; smstufflebeam@mgh.harvard.edu; 4Medical Device & Simulation Laboratory, Massachusetts General Hospital, Harvard Medical School, Cambridge, MA 02139, USA; mottensmeyer@mgh.harvard.edu (M.P.O.); denovi.gianluca@mgh.harvard.edu (G.D.N.)

**Keywords:** MEG, EEG, dipole localization, simulated annealing, global optimization

## Abstract

Dipole localization, a fundamental challenge in electromagnetic source imaging, inherently constitutes an optimization problem aimed at solving the inverse problem of electric current source estimation within the human brain. The accuracy of dipole localization algorithms is contingent upon the complexity of the forward model, often referred to as the head model, and the signal-to-noise ratio (SNR) of measurements. In scenarios characterized by low SNR, often corresponding to deep-seated sources, existing optimization techniques struggle to converge to global minima, thereby leading to the localization of dipoles at erroneous positions, far from their true locations. This study presents a novel hybrid algorithm that combines simulated annealing with the traditional quasi-Newton optimization method, tailored to address the inherent limitations of dipole localization under low-SNR conditions. Using a realistic head model for both electroencephalography (EEG) and magnetoencephalography (MEG), it is demonstrated that this novel hybrid algorithm enables significant improvements of up to 45% in dipole localization accuracy compared to the often-used dipole scanning and gradient descent techniques. Localization improvements are not only found for single dipoles but also in two-dipole-source scenarios, where sources are proximal to each other. The novel methodology presented in this work could be useful in various applications of clinical neuroimaging, particularly in cases where recordings are noisy or sources are located deep within the brain.

## 1. Introduction

The problem of localizing electrical dipole sources as an approximation to localizing true neural activity in the brain plays a crucial role in neuroscience and has many applications in clinical research. Dipole localization techniques based on electroencephalography (EEG) and magnetoencephalography (MEG) use the electrical potentials and magnetic fields generated by underlying neural activity and reaching the scalp to infer the dipole locations in the brain that most likely created the observed data [1,2]. Modeling time-varying scalp-recorded evoked potentials and/or fields emanating from intracranial electrical dipole sources is a complicated problem that relies on three major steps: (a) specification of an accurate volume conduction model of the head and the number and types of intracranial electrical sources, (b) the forward problem of computing electrical and magnetic field intensities over the sensors, given a current source within the brain, and (c) the inverse problem of approximating the current source using the EEG and MEG measurements [3]. Generally, given a suitable source and head model, this inverse problem can be cast as a nonlinear optimization problem of computing the location and moment parameters of the set of dipoles whose projections to the scalp surface matches the EEG and MEG measurements in a least squares sense [4]. In practice, the nonlinear optimization problem can be challenging because its objective function can have many local minima [5], especially at a low signal-to-noise ratio (SNR) [6], which can result from measurement noise or from cases where electrical sources are deep in the brain [7].

To solve the nonlinear optimization problem for dipole localization, current popular optimization techniques can be loosely classified into groups, namely local optimization, grid searches, and global optimization methods [8]. Local search methods include downhill descent methods, such as the quasi-Newton-based gradient descent (GD) and gradient-free search methods, including the Nelder–Mead downhill simplex method [9]. However, these methods are known to be sensitive to the choice of starting parameters, which results in convergence to suboptimal local minima [10]. Dipole scanning (DS) is another widely utilized technique for electrical dipole source estimation that operates on a predetermined grid of precomputed dipole projections [11]. DS systematically identifies the most significant dipole strength at selected time points, determining a dipole’s amplitude, position, and orientation. The accuracy of this approach is heavily contingent upon the sophistication of the head model employed [12], particularly in EEG analysis. Despite its utility, the DS method faces challenges under certain conditions. Particularly in low-SNR environments or scenarios where multiple forward solutions are plausible explanations of the observed data, this technique may lead to dipole localization inaccuracies. In recent years, several different global optimization methods have been proposed for solving the nonlinear optimization problem while being less prone to terminating at local minima. These computational methods include simulated annealing (SA) [13], genetic algorithms (GAs) [14], and the Tabu Search (TS) method [15]. These algorithms avoid local minima by allowing stochastic searches of parameter space but at significantly increased computational cost relative to the local search methods, such as GD and DS.

The issue of GD and DS algorithms giving suboptimal localization results for deep sources, and the use of stochastic search methods to improve upon those results, has been examined by several groups in the past [15,16,17,18]. These prior studies showed that the dipole localization accuracy was significantly improved when using a realistic head model geometry, as compared with single or concentric sphere approximations [19,20,21,22]. Additionally, enhanced localization accuracy by SA and GA was demonstrated when applied to simulated data using a spherical head model, albeit at a high computational cost [6,23]. Several studies have explored the performance of global optimization algorithms, like genetic algorithms (GAs) for electric current source dipole localization [24,25]. For instance, GA has been applied in simultaneous EEG-MEG studies [26] and in analyzing deep sources with low signal-to-noise ratios (SNRs) in the brain [27]. One study compared GA with SA and found that GA was more computationally intensive than SA [28]. Another investigation demonstrated the effectiveness of dipole estimation using SA [18]. Moreover, these prior studies indicated that the combination of concurrently acquired EEG-MEG data achieves superior localization accuracy compared with each modality alone [29] and showed improved source connectivity estimation in EEG and MEG when using realistic head models [30]. Additionally, combined EEG-MEG data were shown to offer complementary information between the two modalities to improve source reconstruction accuracy early in epileptic spike onset, i.e., at time points with a low SNR [31]. Moreover, prior studies [32,33,34,35,36] have highlighted the involvement of deep brain locations in epilepsy generation, emphasizing the importance of accurately identifying the epileptogenic zone and seizure onset zone. However, noninvasive studies often face limitations in achieving precise localization [37]. Recent research has attempted to implement deep learning (DL) methods [8,38] for accurate dipole source localization in EEG and MEG. Despite some progress, these studies are typically constrained to concentric sphere approximations of head models and rely heavily on simulation data. Additionally, they involve computationally intensive calculations [39] and often struggle with low-SNR conditions or when dataset sizes are limited [4,40,41]. 

To mitigate the computational cost of stochastic parameter searches while maintaining the advantage of avoiding local minima, in this work, we propose a novel hybrid SA algorithm, henceforth called hybrid-SA, which uses SA in the initial, broad parameter search, followed by the GD method [26] when SA has converged in the vicinity of a possible global minimum. The performance of the hybrid-SA algorithm was then compared against the commonly used GD and DS [42] algorithms. The dipole localization accuracy of each of these algorithms was tested with simultaneously acquired EEG and MEG data on a realistic pediatric head phantom with embedded dipole sources at different locations and depths from the surface, as has been described previously [43,44]. EEG and MEG dipole data were also simulated using the exact same phantom geometry and different levels of added noise to match the SNR of the physical dipole sources as a function of depth in the head phantom [45]. The virtual phantom simulations enabled studying a wider range of dipole source locations than those available in the physical phantom. For each of the GD, DS, and hybrid-SA algorithms, dipoles were localized, and their orientations were estimated using EEG data alone, MEG data alone, and simultaneously acquired EEG-MEG data that were either physically measured or simulated. The superiority of the hybrid-SA method in low-SNR scenarios was verified for all data types for single and two-dipole source localization scenarios, demonstrating the promise of this approach for future in vivo investigations.

## 2. Methods

### 2.1. Physical Head Phantom

A pediatric human head phantom model based on magnetic resonance imaging (MRI) data of a 3-year-old male child was used. The phantom was molded in successive layers of a conductive polymer, which were cured sequentially, with conductivity values and layer thicknesses that mimicked those of their corresponding head tissues (Table 1) [46,47]. This phantom approximated the human head in that the conductive material simulating the brain tissue was spatially homogeneous, did not differentiate between brain tissue and ventricular spaces, and did not differentiate between dura mater and grey and white matter.

Twisted pairs of electrical wires were installed in hollow cavities that extended from the base of the phantom into the brain volume, reaching different depths from the scalp surface. In total, 12 locations were available for inserting dipole sources, with six sources in each phantom hemisphere. The electrical wires inserted in those 12 locations each ended with tips that had gaps in the 0.8–1.7 mm range, so that artificial electromagnetic dipoles were generated when time-varying voltages were applied, simulating neuronal pulse sources at the distal ends. Each source location, in Cartesian (x, y, z) coordinates, and dipole source orientation, represented by the azimuthal angle Theta (θ) and the polar angle Phi (φ) in the spherical coordinate system, was determined using a 3D computed tomography (CT) image volume of the head phantom. Each dipole vector **d** was defined from one wire tip to the other as $\left(x_{2}-x_{1},y_{2}-y_{1},z_{2}-z_{1}\right).$ The azimuthal angle θ, representing the angle in the xy-plane from the positive X-axis, was calculated using the arctangent function, specifically $\theta =atan2(y_{2}-y_{1},x_{2}-x_{1})$. The polar angle φ, measured from the positive Z-axis towards the x-y plane, was derived using the arccosine of the ratio of the z-component of the dipole vector to its magnitude $\emptyset =acos\frac{z_{2}-z_{1}}{\surd {{(x}_{2}-x_{1})}^{2}+{{(y}_{2}-y_{1})}^{2}+{{(z}_{2}-z_{1})}^{2}}$. These calculations provided the necessary spherical coordinates to accurately model the orientation of each dipole in relation to the head phantom’s coordinate system. Specifically, the spatial coordinates designating the true dipole locations were determined by computing the midpoint along the x-, y-, and z-axes of the two exposed wire tip-ends, creating a dipole at the end of each wire, as derived from the head phantom CT scan (Table 2).

A voltage generator (RIGOL DG1032z, China) was used to play back signals through one dipole at a time. The waveforms were based on epileptic spikes of an actual human pediatric subject, as previously classified by an expert. These signals were recorded simultaneously using a combined EEG-MEG setup. Simultaneous MEG and HD-EEG recordings were performed at the MEG facility of Cook Children’s Medical Center (Fort Worth, TX, USA). The recordings were performed in a one-layer magnetically shielded room (Imedco, Hägendorf, Switzerland) with a whole-head Neuromag^®^ Triux 306-sensor system (MEGIN, Helsinki, Finland). HD-EEG was recorded using a MEG-compatible 256-channel Geodesic HD-EEG system (Magstim-EGI, OR, USA). Standard co-registration procedures were followed for MEG and HD-EEG according to the manufacturer’s instructions. SNR was defined as the proportion of the desired neural signal relative to background noise, indicating the clarity with which brain’s electromagnetic activity of interest is distinguishable from other interfering signals [48]. The SNR of each of the EEG and MEG recorded signals was determined as the ratio of the maximum activation to the baseline standard deviation. The baseline period, defined as the interval from −250 ms to −10 ms prior to the onset of each spike event [49], served as a reference for pre-stimulus neural activity. The exclusion of interval from −10 ms to 0 ms in the baseline period eliminated any potential pre-spike activity that could bias the baseline measurement, ensuring a more reliable comparison between the desired neural signal and background noise.

### 2.2. Computational Head Phantom

Based on the T1-weighted MRI of the same pediatric subject, a computational model of the virtual head was created in Fieldtrip [50]. The MRI voxels were segmented into the three different tissue types, using the *ft_volumesegment* function: scalp, skull, and brain [51]. Following segmentation, triangulated meshes were constructed that described the tissue surface boundaries using *ft_prepare_mesh* [52]. The tissues from which the surfaces were created and the number of vertices for each tissue were specified (3000, 2000, 1000, for scalp, skull, and brain, respectively). Since the brain tissue was assumed to be homogeneous, most of the attention on setting up the boundary element solution for dipole sources focused on the density of surface meshes. To that end, the commonly used ratio of 3/2/1 for scalp/skull/brain was implemented.

Once the scalp, skull, and brain had been segmented and surface descriptions were constructed for each, the *ft_prepare_headmodel* function was used to create the volume conduction model for the head phantom. Then, the OpenMEEG [53] method was used to generate a 3-layer BEM-based head model for electromagnetic wave propagation. The relative conductivity values, normalized to scalp conductivity using the values in Table 1, were then specified in OpenMEEG as [1, 0.0125, 1] for scalp (=head), skull (=outer skull), and brain (=inner skull), respectively.

In addition to the physical phantom recordings for EEG and MEG data for activated sources, as shown in Figure 1, several other intracranial dipoles were simulated at various locations and depths of a virtual phantom of identical geometry, as shown in Figure 2, to increase the density of the data to be analyzed by each algorithm (Table 3). The locations were carefully chosen to span source depths ranging from 0.5 cm to 4.5 cm beneath the scalp, ensuring coverage of both superficial and deep-seated brain regions that are typically capable of generating spike activity [32,34,54].

### 2.3. EEG and MEG Data Analyses

For the physical phantom measurements, the simultaneous EEG-MEG data were first pre-processed. The acquired EEG and MEG for the same source data were merged into a single fif file using MATLAB. To remove noise from the signals, the channels that exhibited artifacts or high-amplitude noise (≥5x the mean value of the signal strength of all the channels) were marked as bad channels and were excluded from the analyses. Filtering was then performed on the data (band-pass filter: 1–70 Hz, notch filters: 60 Hz and multiples up to the fourth harmonic) [55]. Given that this study utilized a human head phantom, spatial filtering techniques such as SSP or ICA [56], generally employed to identify and mitigate sensor topographies associated with specific artifacts such as heartbeats and eyeblinks, were not necessary. The spikes observed in the EEG and MEG recordings were identified as spike events, and the potentials/fields epoched around these spikes [42] were used to model the electrical dipoles for EEG, MEG, and simultaneous EEG-MEG data.

For the computationally generated measurements, the noise model for simulating varying SNR conditions in relation to source depths was developed using real EEG-MEG measurement data. The SNR values for EEG-MEG measurements for source activations at different depths were first computed from experimentally measured phantom data. To determine the rate at which noise increases with depth, the SNR values were converted to a logarithmic scale, allowing for a linear relationship to be established. The log-transformed SNR values were fitted to depth using linear regression, yielding a slope. The positive constant, which governed the rate of exponential increase in noise with depth, was then derived from the slope using the following relation:(1)$$\beta =\frac{-slope}{2}$$

Gaussian noise was added to the simulated sources in such a manner that deep-seated sources had lower SNR values compared to the superficial sources, to match the SNR of the experimentally measured phantom data as a function of dipole depth.
(2)$$Noise\left(d\right)=N(0,\sigma^{\beta d})$$
where σ is the baseline standard deviation of the noise and β is a positive constant determining the rate at which the standard deviation of the noise increases with depth *d*.

For this study, the baseline level of noise, determined by the ratio of sigma to signal amplitude, was set to 0.1.

For both physical and computationally generated measurements, the SA algorithm (starting temperature: 150, cooling rate: 0.95) was run until its termination criteria were met (optimality tolerance < 1.0 × 10^−5^, or 10,000 iterations were reached). The final output parameters then served as the starting point for the GD algorithm to further optimize the final local search. For our hybrid-SA algorithm, the relative weight assigned to the EEG and MEG data in the penalty function was defined by:(3)$$P=\lambda \;\times \;{\mathrm{SNR}}_{\mathrm{EEG}}\;\times \;\sum_{i=1}^{NEEG}{\left(O_{EEG},i-M_{EEG},i(p)\right)}^{2}+(1-\lambda )\;\times \;{\mathrm{SNR}}_{\mathrm{MEG}}\;\times \;\sum_{j=1}^{NMEG}{\left(O_{MEG},j-M_{MEG},j(p)\right)}^{2}$$
where λ = $\frac{{GOF}_{EEG}}{{GOF}_{EEG}+{GOF}_{MEG}}$.

The λ parameter served as a weighting factor to balance the relative contributions of EEG and MEG data in the penalty function. NEEG and NMEG represented the total number of EEG and MEG data points, respectively. OEEG,i and OMEG,j were the observed data points for EEG and MEG. MEEG,i(p) and MMEG,j(p) were the predicted values from the model for EEG and MEG data points, based on the set of parameters p, which included dipole positions and orientations. The sums calculated the squared differences between the observed and predicted values for each modality and provided the value of the metric to be minimized to achieve the best possible model fit to the combined EEG-MEG data. For the cases where EEG-only or MEG-only data were being fitted, the value of λ was set to 1 and 0, respectively.

The goodness of fit (GOF) was computed as 1-residual variance, where residual variance is as follows:(4)$$\frac{\Sigma {\left(d_{1}-d_{2}\right)}^{2}}{\Sigma {d_{1}}^{2}}$$
with d_1_ representing the actual EEG/MEG recording and d_2_ representing the simulated recording for a given dipole location and orientation when dipole placement occurs during dipole fitting. The sum over the squared differences between the actual EEG/MEG recordings (d_1_) and the simulated recordings (d_2_) across all channels, as expressed by $\Sigma {\left(d_{1}-d_{2}\right)}^{2}$, gave the cumulative measure of model’s error. This total squared discrepancy was then normalized by the sum of the squares of the actual recordings for each EEG/MEG channel, providing a relative measure of the model’s error compared to the inherent variance in the data. This formulation quantified the fidelity of the fitted data to the actual recordings by presenting the unexplained variance as a fraction of the total variance in the observed data, thereby directly assessing the model’s accuracy.

### 2.4. Statistical Analyses of Dipole Localization Algorithm Parameter Estimates

Lastly, the performance of each algorithm was quantified by the Euclidean distance [57] between the actual (CT-derived and simulated) source and the fitted dipole locations and correspondingly for the dipole azimuthal and polar angle differences [58]. For the case of two-dipole localization measurements, the data were created synthetically by adding two datasets from the corresponding single-dipole data obtained experimentally or computationally. The two-dipole localization performance was then evaluated based on the averages of the Euclidian distance errors for the two sources.

For the statistical analyses of dipole localization and orientation errors, a total of 260 spikes were generated from each source, which were organized into 13 groups of 20 spikes each. These spikes were recorded over a period of just over 3.5 min at a sampling rate of 2000 Hz. For each group, the dipole error was computed by fitting dipoles to the averaged peak values of the 20 spikes. Given the non-normal distribution of the resulting dipole error data, a robust statistical approach was employed involving the Friedman test [59], a non-parametric alternative to repeated measures ANOVA. This test was utilized to evaluate the statistical significance of variations among the dipole errors associated with three algorithms—GD, DS, and hybrid-SA. When the Friedman test showed a statistically significant difference among algorithms (*p* < 0.05), we then tested all pairwise comparisons between algorithms while addressing non-normality and potential Type I errors, using the Tukey–Kramer post hoc test.

## 3. Results

An intuitive explanation of the superior dipole localization results arrived at by the hybrid-SA algorithm is depicted in Figure 3. Figure 3a shows a cross-section of the GOF space along the x-y plane for the z location of source Right 6, which was deeper than most other sources and had an EEG SNR of 3.6. The figure shows that the global minimum arrived at by the hybrid-SA algorithm was in a significantly different location from the GD algorithm. In contrast, Figure 3b shows corresponding results for the more superficial source Right 4, which had an EEG SNR of 7.1. In this case, the local minimum found by the GD algorithm was within a few mm from the global minimum found by the hybrid-SA algorithm, so an improvement was still attained by using the latter algorithm, but it was smaller. For completeness, it should be mentioned that all dipole localization searches were conducted in three spatial dimensions (x, y, z) and two angle dimensions (θ, φ), but only a 2D example of the GOF space is shown here for visualization purposes.

Single-dipole source localization error results for the three optimizations algorithms, compiling both the physical and simulated phantom EEG data, are shown in Figure 4.

Figure 4a shows that the hybrid-SA algorithm outperforms the DS and GD algorithms consistently, with a statistically significant localization improvement starting from an SNR value ~6.1 but more so at lower SNRs. In the latter case, the localization improvement was up to 20 mm, though the remaining error was still large, in the 20–25 mm range. It appears that the maximum benefit obtained from the hybrid-SA algorithm for accurate localization of dipoles, i.e., within 10–15 mm of their true location, from EEG data was for measurements in the 4–6 SNR range. In this medium SNR range, the goodness-of-fit (GoF) space is characterized by multiple peaks, making it likely for the GD and DS algorithms to become stuck in local minima. The hybrid-SA algorithm, however, can escape these local minima, leading to more accurate localization. This highlights the algorithm’s robustness in challenging conditions with low SNRs, a critical advantage for practical applications. Figure 4b shows results from the same data analyses but plotting localization errors as a function of dipole source depth. While the average performance of the DS method demonstrated a trend for better results than the GD method for most depths, no statistically significant differences (*p* < 0.05) were observed between the two methods at any source depth. In contrast, the hybrid-SA algorithm consistently demonstrated lower errors compared to the other two techniques, with a statistically significant localization improvement over the other algorithms at depths exceeding 14 mm from the surface of the cortex. These findings suggest that the hybrid-SA algorithm is particularly effective for deeper sources, providing a reliable solution where other methods fall short. Additionally, none of the algorithms converged for SNR < 2.5, underscoring the limitations in extremely low-SNR scenarios.

Figure 5 shows the corresponding localization error results when using MEG data alone, which were qualitatively similar to those found above for EEG data. Figure 5a shows increasingly improved localization for the hybrid-SA algorithm with lower SNR down to ~3, beyond which point no algorithm converged. Interestingly, improvements in localization error for the lowest SNR values were higher for MEG (error < 20 mm) than EEG (error < 25 mm) data. However, SNR values in the 4–6 range were still required for accurate localization, defined as errors < 15 mm. Figure 5b shows corresponding localization errors as a function of dipole depth. Interestingly, there was a statistically significant improvement with the hybrid-SA method over the DS and GD methods at depths exceeding 10 mm from the scalp surface, corresponding to relatively high SNR values (<~7). There was also a statistically significant difference between the DS and GD methods, with the DS method outperforming GD at depths exceeding 30 mm from the surface of the head, corresponding to SNR < ~4. Furthermore, it was observed for the same source in the phantom that the MEG SNR was higher than the EEG SNR at greater depths, which likely contributed to the observed differences in localization accuracy between the two modalities. Like the EEG modality, here too, the hybrid-SA algorithm demonstrated the best performance within an SNR range of 4 to 6.

In the next step of these analyses, synchronized EEG and MEG data were used simultaneously for dipole localization (Figure 6). Figure 6a shows that all three algorithms performed better compared to their own corresponding performances for EEG data alone and MEG data alone. There was up to a 25% improvement in source localization accuracy for the hybrid-SA algorithm for simultaneous EEG-MEG compared to EEG or MEG data alone. Importantly, even for the lowest usable SNR of ~3, the localization error did not exceed ~18 mm, demonstrating the potential of this approach for improved deep-source localization. Similarly, there was up to a 35% improvement in the localization accuracies with GD and DS using the combined EEG-MEG data compared to their own performance when using EEG or MEG data alone. Importantly though, the hybrid-SA algorithm significantly outperformed the other methods at a lower SNR, by up to 50% in localization accuracy. Figure 6b shows the corresponding localization error results as a function of dipole depth. For depths exceeding 10 mm, there was a statistically significant difference between the localization errors for the hybrid-SA algorithm and GD. Similarly, for depths exceeding 17 mm, the hybrid-SA algorithm outperformed the DS method. Although DS showed a trend that hinted that its performance was a little better than GD, no statistically significant differences were found between DS and GD in pairwise comparisons at any source depth.

Subsequently, comparative analyses of mean squared localization errors were performed for all possible pairwise combinations of two simultaneously activated dipole sources for both simulated and physical source data. Figure 7 shows that, when compared to the GD and DS methods, the hybrid-SA algorithm had up to a ~50% improvement in localization accuracy for EEG data alone (Figure 7a), up to a ~40% improvement for MEG data alone (Figure 7b), and up to a ~45% improvement when using combined EEG-MEG data (Figure 7c). Interestingly, when localizing two dipoles, the total error decreased for all data analyses, especially at lower SNRs, relative to single-dipole localization. For example, at SNR ~3, the EEG-only error was reduced from 26 mm (Figure 4a) to 22 mm (Figure 7a), the MEG-only data error was reduced from 20 mm (Figure 5a) to 17 mm (Figure 7b), and the combined EEG-MEG data error was reduced from 18 mm (Figure 6a) to 14 mm (Figure 7c).

Lastly, it should be noted that although the azimuthal and polar angles of each dipole in all analyses were varied freely in addition to their three-dimensional coordinates, the improvements brought about by the hybrid-SA algorithm versus GD algorithm were significantly better at all SNRs; however, the improvements versus DS were not as significant as they were for dipole localization. Figure 8a shows that for combined EEG-MEG data, the relative improvement in azimuthal-angle determination was about 5 degrees for all SNR values, even though the absolute error was as little as 5 degrees at high SNRs. Similarly, Figure 5b shows that the relative improvement in polar-angle errors was about 10 degrees for all SNR values, and the absolute error was as little 5 degrees in absolute terms, also for the highest SNRs. Qualitatively similar results were also obtained for EEG data alone and MEG data alone, with MEG data performing slightly better than EEG at all SNR values (Appendix A).

## 4. Discussion

Conventional GD algorithms used in dipole fitting rely on the gradient (first-order derivative) function to navigate through the error landscape [60], which can lead to trapping at local minima. DS is another popular technique for electrical dipole-source estimation that operates on a predetermined grid of precomputed dipoles. The accuracy of this approach is heavily contingent upon the complexity of the head model employed, particularly in EEG analysis. The inherent limitation of working within a fixed grid of dipoles is that it can sometimes hinder the ability to accurately capture dipole localization in fine-enough detail due to the excessive computational cost that this would entail. In contrast, the SA method offers a probabilistic approach for approximating the global optimum of a function, which is computationally parsimonious [61]. Initially, the SA method may accept suboptimal solutions, but as the iterations progress, the likelihood of selecting inferior solutions diminishes [13,17,62]. As a result, at lower annealing temperatures, the SA search is constrained to a local minimum, although this search remains stochastic in nature. To expedite the global optimization process, once the annealing temperature is likely to have led the search near a local minimum, we implemented a hybrid-SA method. Once an empirically determined iteration number criterion was met, the SA algorithm was stopped and an accelerated search towards the nearest minimum was completed by the GD algorithm.

The overall results of this work show that the hybrid-SA method was significantly more robust in converging to the global minima compared to the GD and DS methods, especially at lower SNRs (~3–6) for all three modalities. Figure 3b illustrates that the optimization landscape is more rugged at lower SNRs, which increased the susceptibility of GD and DS methods to being stuck at local minima. Interestingly, even at higher SNRs (~7–9), the hybrid-SA method retained an advantage, albeit smaller, because the error landscape remained somewhat rugged, making local and global searches land at nearby, but not identical, minima (Figure 3a). Although Figure 3 shows an example of the two-dimensional dipole-source localization error, qualitatively similar landscapes were created for all three spatial dimensions and the two angle dimensions (Appendix A).

Source localization errors were compared for EEG or MEG data alone and for combined EEG-MEG data. The purpose of this analysis was to test whether the hybrid-SA algorithm (1) achieved more accurate source localization for EEG-only data than MEG-only data for deeper source locations, as magnetic fields are known to decay faster than electric fields as a function distance from their dipole source [63], (2) achieved more accurate source localization for MEG-only data than EEG-only data for more superficial source locations for similar physical reasons [64], and (3) improved source localization when using the combined MEG-EEG data due to the complementary strengths of the two modalities [29,53,54].

For EEG-only data, in scenarios with a high SNR (>7), all three algorithms demonstrated localization errors under 5 mm. However, as the source depth increased and SNR decreased (Figure 4b), the dependence of localization error on SNR could be approximated by a negative exponential trend, as evidenced by an exponential trendline fitted across all data points in Figure 4a. The SNR decrease resulted in an ever-increasing advantage for the hybrid-SA algorithm, resulting in a notable ~45% improvement in dipole localization accuracy compared to the other two methods. However, at the lowest measurable SNR of ~3, the absolute localization error was ~26 mm, which was still large.

For MEG-only data, a slightly higher SNR than EEG was required to not exceed a source localization error of 5 mm (SNR ~ 7.7). However, the hybrid-SA algorithm attained significantly improved localization for all source depths (depth > 10 mm, Figure 5b), which was comparable to that of EEG-only data (depth > 14 mm, Figure 3b). At low SNR values, on the other hand, source localization was superior for the same sources localized by EEG data alone, by all three algorithms. At the lowest usable SNR of ~3, the hybrid-SA localization error for MEG-only data was ~20 mm (Figure 5a) versus ~26 mm for EEG-only data (Figure 4a). This finding was surprising given that detected MEG signals are known to decay faster with increasing tissue depth [55,56]. However, our phantom lacked background electrical activity from other neuronal sources, as is typical for in vivo measurements, and this likely biased results in favor of the MEG measurements.

The combination of MEG-EEG data (Figure 6) resulted in a higher SNR and, thus, better localization accuracy, compared to the individual EEG or MEG modalities alone, for all three algorithms. The localization error followed a negative exponential trend, as before, but the exponent value was steeper for the combined data (Figure 6a) versus the MEG-only (Figure 5a) or the EEG-only (Figure 4a) data, the latter having the smallest negative exponent. As a result, the MEG-EEG data attained a localization error of ~18 mm, even at the lowest usable SNR of ~3. Interestingly, the improved SNR for the combined MEG-EEG data resulted in the hybrid-SA algorithm starting to outperform the DS algorithms at slightly greater depths (depth > 17.1 mm; Figure 6b) compared to MEG and EEG data alone (Figure 4b and Figure 5b, respectively). Given that the phantom used in this study was a pediatric one, these data suggest that when combing MEG with EEG data, local optimization algorithms like GD and DS can still localize sources as well as the more computationally expensive hybrid-SA algorithm within the first ~2 mm in the child brain, i.e., near the cortical surface, but the increased computational cost of the hybrid-SA algorithm pays dividends at greater depths.

In the last part of this study, the three algorithms were tested for MEG and EEG data generated by two simultaneously activated dipole sources (Figure 7). Consistent with expectations, the highest localization accuracy was observed for the combined MEG-EEG data. Our findings revealed a notable improvement in the performance of all three algorithms when using the two-dipole models as a function of SNR for EEG-only (Figure 7a), MEG-only (Figure 7b), and combined MEG-EEG data (Figure 7c). This accuracy enhancement compared to fitting single-dipole sources can be attributed to the more constrained parameter search space for a two-dipole model when searching for specific dipole orientations and amplitudes [65]. Although all algorithms showed improved localization when fitting the two sources with coincident activation, the hybrid-SA algorithm showed the most impressive performance, attaining a ~14 mm localization error, even at SNR ~3.

Lastly, in addition to spatial localization errors, comparisons were performed for all the above-discussed scenarios for dipole-angle errors in theta and phi. These analyses demonstrated similar patterns in angle errors (Figure 8) across all three modalities (EEG, MEG, and combined EEG-MEG). The improvements made by the hybrid-SA algorithm for the dipole angles were statistically significant versus GD at all SNRs, and while the improvement over DS was not as dramatic as it was for the spatial localization, the overall picture was that the combined EEG data reduced the absolute error for both theta and phi to ~5° for the highest SNR values of 8–9. Even for a moderate SNR of 4–5, the errors in both angles barely exceeded ~10°.

## 5. Conclusions

This work has demonstrated the promise of hybrid-SA, a global optimization algorithm combined with an accelerated local GD search, to outperform the often-used GD and DS algorithms for dipole localization, especially at a low SNR. Future work could involve more sophisticated phantom studies that model interference from background dipole sources competing with the target dipole source, to help these measurements approximate in vivo conditions more closely. Additionally, an advantage of combining MEG with EEG data is that the latter can see both radial and tangential sources, whereas the former only sees tangential ones [66]. However, this distinction could not be tested with the phantom used in this study, as none of the dipole sources used were purely radial. In future work, phantoms, including some purely tangential and some purely radial sources, will be tested. With more simulated sources in the phantom at different depths with different SNRs, the hybrid-SA-algorithm-optimized deep learning method for dipole estimation can also be studied [67]. Finally, more work is needed in the future to test the hybrid-SA algorithm performance for different two-dipole scenarios in phantom-based measurements and for in vivo data. Possible scenarios may include the coincident activation of two sources that do not have large spatial separation in a subject’s brain, e.g., they are on the same hemisphere or brain sub-region. Another scenario where the hybrid-SA algorithm could help could involve its use with a responsive neurostimulation (RNS) device in a phantom-based, or an in vivo, study, where it could potentially compensate for the RNS device-induced distortions to improve localization of the epileptic area in the brain.

## Figures and Tables

**Figure 1 bioengineering-11-00897-f001:**
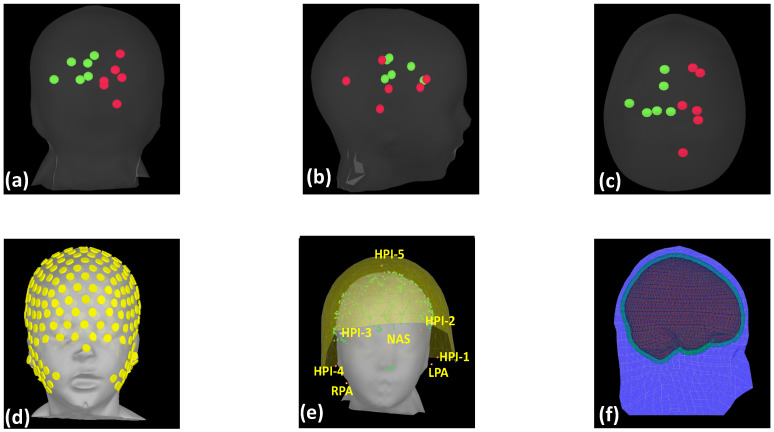
(**a**–**c**) Coronal, sagittal, axial views, respectively, of the dipole sources inside the phantom model. Green represents the sources on the left hemisphere and red represents the sources in the right hemisphere, six per hemisphere. (**d**,**e**) EEG and MEG sensor arrangements, respectively, for the phantom measurements. (**f**) A realistic BEM-based head model for projecting the dipole sources onto the scalp surface.

**Figure 2 bioengineering-11-00897-f002:**
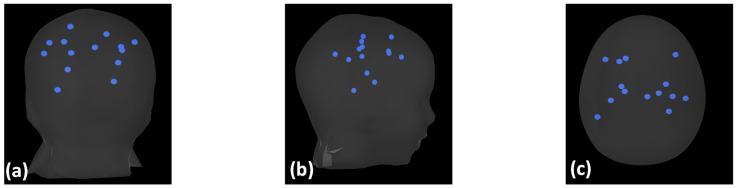
(**a**–**c**) Coronal, sagittal, axial views, respectively, of the simulated sources inside the phantom model with blue spheres representing the locations of the simulated sources.

**Figure 3 bioengineering-11-00897-f003:**
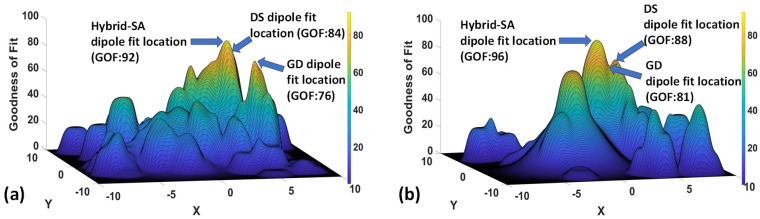
Goodness of fit (GOF) for the localization of an EEG dipole source as a function of source location across an axial plane of the phantom ((**a**) z_1_ = 33 mm, (**b**) z_2_ = 29 mm).

**Figure 4 bioengineering-11-00897-f004:**
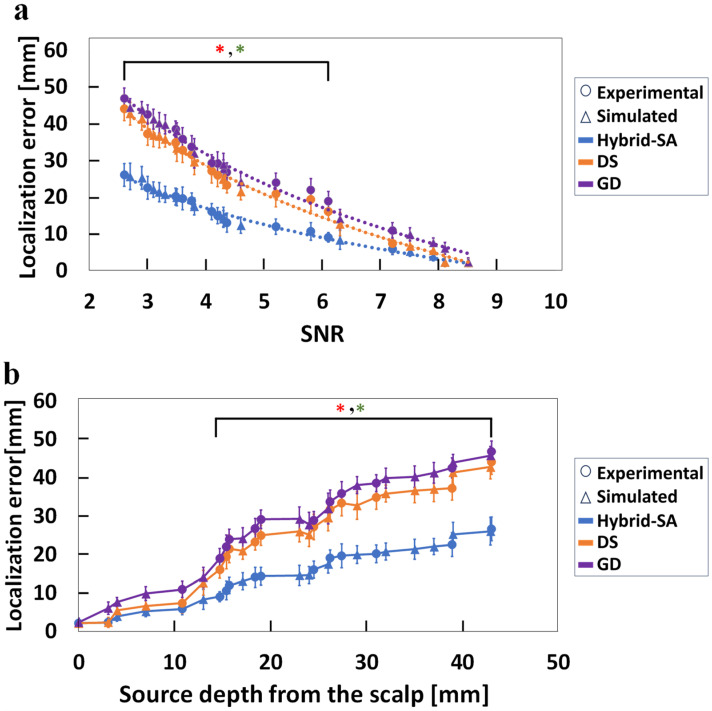
Comparison of the three localization techniques, hybrid-SA, DS, and GD, for (**a**) localization error versus SNR, and (**b**) localization errors versus dipole depth, when using EEG data only. Simulated and experimental data are plotted together, but with different symbols. (* representing statistically significant difference between hybrid-SA and GD, * representing statistically significant difference between hybrid-SA and DS, for *p* < 0.05).

**Figure 5 bioengineering-11-00897-f005:**
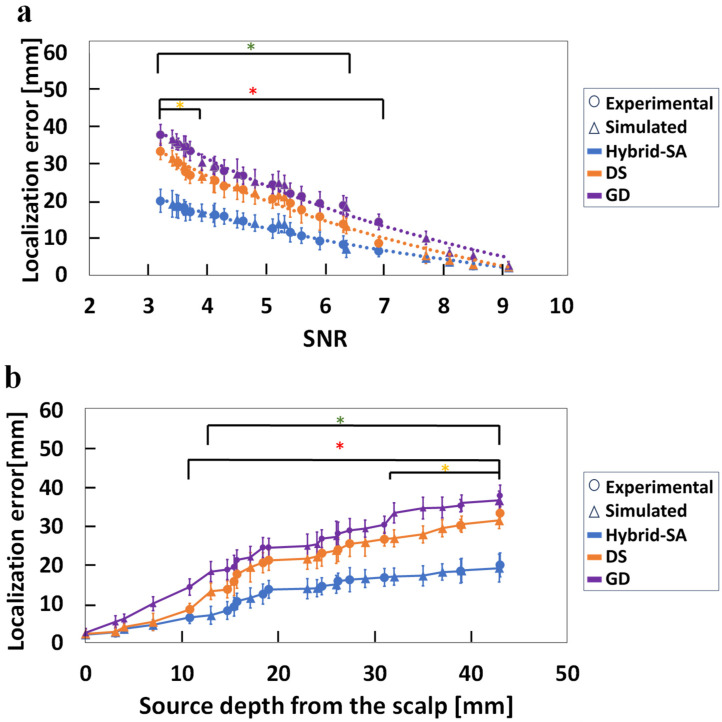
Comparison of the three localization techniques, hybrid-SA, DS, and GD, for (**a**) localization error versus SNR, and (**b**) localization errors versus dipole depth, when using MEG data only. Simulated and experimental data are plotted together, but with different symbols. (* representing statistically significant difference between hybrid-SA and GD, * representing statistically significant difference between hybrid-SA and DS, * representing statistically significant difference between GD and DS for *p* < 0.05).

**Figure 6 bioengineering-11-00897-f006:**
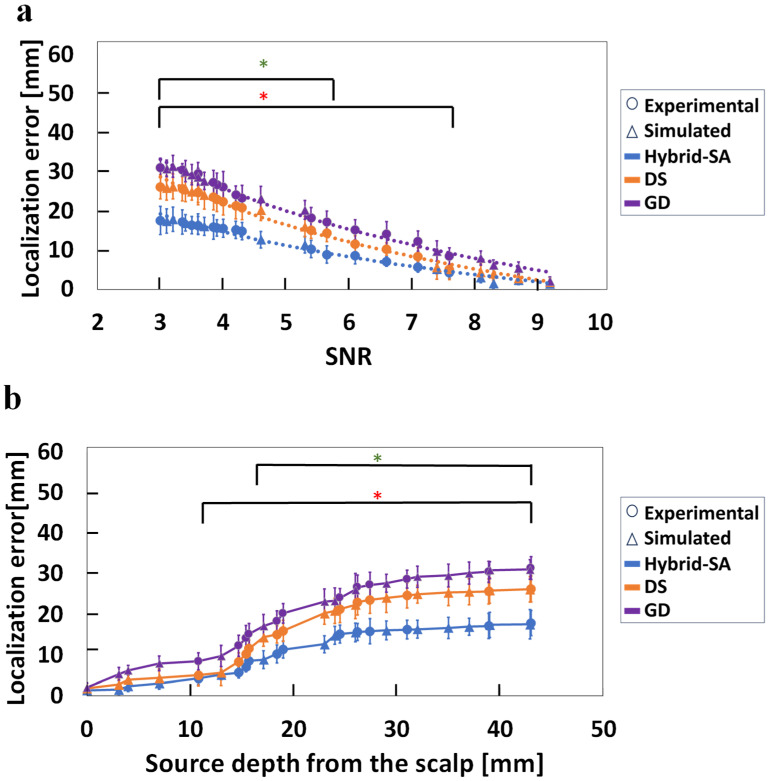
Comparison of the three localization techniques, hybrid-SA, DS, and GD, for (**a**) localization error versus SNR, and (**b**) localization errors versus dipole depth, when using the combined MEG-EEG data. Simulated and experimental data are plotted together, but with different symbols. (* representing statistically significant difference between hybrid-SA and GD, * representing statistically significant difference between hybrid-SA and DS, for *p* < 0.05).

**Figure 7 bioengineering-11-00897-f007:**
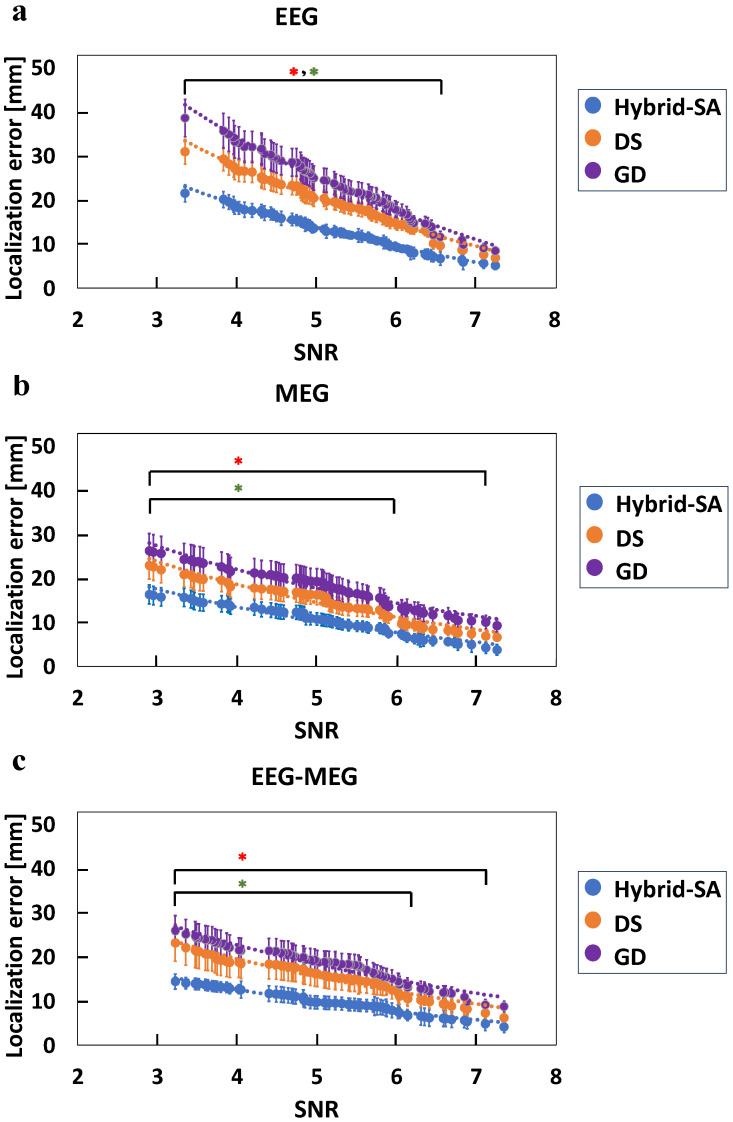
Comparison between the three localization techniques, hybrid-SA, DS, and GD for the simultaneous localization two dipoles for (**a**) EEG-only, (**b**) MEG-only, and (**c**) combined EEG-MEG data, versus SNR. (* representing statistically significant difference between hybrid-SA and GD, * representing statistically significant difference between hybrid-SA and DS, for *p* < 0.05).

**Figure 8 bioengineering-11-00897-f008:**
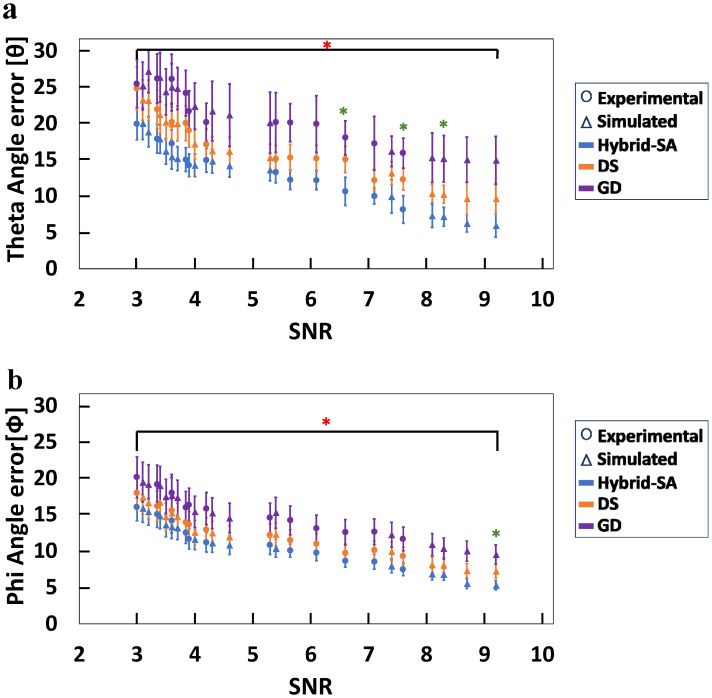
Comparison between the three algorithms, hybrid-SA, DS, and GD for (**a**) dipole phi (azimuthal) angle error, and (**b**) dipole theta (polar) angle error versus SNR. (* representing statistically significant difference between hybrid-SA and GD, * representing statistically significant difference between hybrid-SA and DS, for *p* < 0.05).

**Table 1 bioengineering-11-00897-t001:** Thickness and conductivities of the layers of the pediatric MRI-based pediatric phantom model.

Tissue Type	Tissue Layer Thickness in mm	Conductivity in S/m
Brain	2.76 mm	0.330
Skull	4.16 mm	0.004
Scalp	3.90 mm	0.330

**Table 2 bioengineering-11-00897-t002:** Locations and orientations of each dipole source in the head phantom as determined by CT.

Source	[x y z] Location in mm in SCS System	Theta (θ) Value in Degrees	Phi (φ) Value in Degrees
Right 1	[8.62, −21.4, 0.63]	112	36
Right 2	[−36.76, −5.14, 45.82]	68	186
Right 3	[−0.27, −18.41, 12.01]	31	107
Right 4	[10.66, −5.30, 28.83]	47	78
Right 5	[56.09, −16.60, 33.42]	48	112
Right 6	[47.54, −23.29, 27.32]	135	84
Left 1	[13.39, 27.66, 53.16]	112	26
Left 2	[7.68, 18.73, 35.37]	32	121
Left 3	[16.92, 4.60, 59.22]	36	48
Left 4	[15.76, 44.49, 33.32]	68	53
Left 5	[40.31, 11.13, 44.68]	73	191
Left 6	[52.03, 10.55, 27.39]	116	118

**Table 3 bioengineering-11-00897-t003:** Locations and orientations of the dipole sources simulated at various depths in the virtual head phantom.

Source	[x, y, z] Location in mm in SCS System	Theta (θ) Value in Degrees	Phi (φ) Value in Degrees
S1	[50.68, 26.54, 65.3]	108	78
S2	[55.24, 20.18, 82.11]	56	113
S3	[53.44, 41.38, 63.12]	24	57
S4	[−15.87, 51.26, 69.33]	124	63
S5	[5.30, −43.02, 78.15]	146	119
S6	[12.60, −14.18, 85.23]	171	138
S7	[7.65, −29.41, 71.78]	21	54
S8	[57.01, −33.96, 54.12]	59	78
S9	[14.14, 21.36, 62.13]	112	91
S10	[1.21, 34.98, 21.68]	74	162
S11	[4.41, 41.26, 44.73]	82	49
S12	[−11.27, −25.57, 58.12]	61	36
S13	[29.12, −23.44, 45.47]	38	87
S14	[7.77, −2.51, 70.69]	45	76

## Data Availability

Data supporting the results of this study are available from the corresponding author upon request.

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
