# Peer review of "Improved Dipole Source Localization from Simultaneous MEG-EEG Data by Combining a Global Optimization Algorithm with a Local Parameter Search: A Brain Phantom Study"

_bioengineering, 2024, doi:10.3390/bioengineering11090897_

Round 1

Reviewer 1 Report

Comments and Suggestions for Authors

The idea conceived by the authors was fine and good.

What is meant by signal to noise ratio? no explanation.

Literature survey was not sufficient.

Any specific reason for selecting the position ?

Baseline period was defined as the interval from -250 ms to -10 ms why?

How did you remove noises from the signals?

Need more explanation in the result and analysis part.

Author Response

Dear Sir/madam,

Thank you for your comments and questions. We believe that your feedback has significantly contributed to improving the quality of our work.

Please find attached  our detailed responses to each comment and question raised. We have addressed all the points raised and incorporated the necessary changes to enhance the clarity and depth of our manuscript.

With regards,
Subrat Bastola

PhD Student
University of Texas at Arlington

Reviewer 2 Report

Comments and Suggestions for Authors

This study introduces a hybrid optimization algorithm combining Simulated Annealing and Gradient Descent for improved dipole source localization using simultaneous EEG and MEG data. The manuscript demonstrates significant improvements in localization accuracy, especially in low SNR conditions and for deep brain sources, compared to traditional methods. The study's findings suggest potential applications in clinical neuroimaging, with future work needed to validate the algorithm in more complex and realistic scenarios.

There are some notable strengths of the manuscript,

1.     The hybrid-SA algorithm represents an approach to addressing challenges in dipole localization, particularly for low SNR and deep brain sources.

2.     The paper provides a thorough comparison of the hybrid-SA algorithm with existing techniques, offering insights into its strengths and limitations.

3.     The study has potential for improving dipole localization in clinical neuroimaging, which could enhance diagnostic accuracy and patient outcomes.

4.     The study uses both physical and computational head phantoms, providing a evaluation framework for the proposed algorithm.

Few Questions / Comments on the manuscript are as follows,

1.     Figure 1. Does show mismatch in sub-figures a, b, and c. It mentions that the figures shows coronal, sagittal, and axial images, whereas the description says it is Sagittal, axial and coronal respectively. It would be good to correct the order of the images for the dipole sources inside the phantom model.

2.      Similarly for figure 2, the figure description and the figures doesn’t align. The order of figure seems to be coronal, sagittal and axial whereas the description says it is Sagittal, axial and coronal respectively.

3.     The manuscript introduces a hybrid Simulated Annealing (SA) and gradient descent (GD) algorithm for dipole localization in MEG-EEG data. How does this approach fundamentally differ from other existing hybrid methods in this space, and what is the specific novel contribution to the field?

4.     In this study, the objective of combining SA with GD is to improve localization accuracy in low SNR scenarios. Could you elaborate more on why SA was chosen as the global optimization method? Were other global optimization methods like Genetic Algorithms (GA) considered, and if so, why were they not selected?

5.     In this paper, section 2.3, for EEG and MEG data analyses, mentions the use of simulated EEG-MEG data. Can you provide more details on the noise model used for simulating low SNR conditions? How was the Gaussian noise level calibrated to match real-world conditions?

6.     The manuscript describes the combination of SA and GD of the proposed method. Could you clarify how the transition between SA and GD is managed? Is there a specific criterion or threshold that triggers the switch from the global to the local search? Also, how did the author decide on the termination criteria?

7.     Also, could the author describe on the sequence of models, the author tested the sequence of using the SA model 1st and then the GD model. Was there a specific reason to select this sequence? Also, was there any alternative approach tried with the combination of models?

8.     Section 2.3, for EEG and MEG data analyses, in the preprocessing step, channels with artifacts were excluded. How does this exclusion impact the overall dataset, especially in terms of signal integrity and noise levels? Were alternative methods considered for dealing with artifacts, such as signal reconstruction?

9. The manuscript compares the hybrid-SA method with the GD and DS methods. How does it perform relative to more recent deep learning-based methods for dipole localization?

Author Response

Dear Sir /Madam,

Thank you for your comments on the manuscript.We believe that your feedback has significantly contributed to improving the quality of our work.

Please find attached the revised manuscript along with our detailed responses to each comment and question raised . We have addressed all the points raised and incorporated the necessary changes to enhance the clarity and depth of our manuscript.

With regards,

Subrat Bastola

PhD Student

University of Texas at Arlington
